# Learning to Look Harder: Position-Aware Attention Intensity Modulation in Transformers

## Abstract

Transformer attention mechanisms waste computational resources by applying uniform intensity across all sequence positions, treating simple and complex contexts equally. We propose attention intensity modulation, a lightweight method that dynamically scales attention strength through multi-head position-aware complexity prediction. Our approach augments each attention block with a predictor that outputs head-specific intensity factors (0.2–1.0), scaling attention scores before softmax based on both content embeddings and learned positional information. Comprehensive evaluation across four text modeling datasets (shakespeare_char, enwik8, text8, Project Gutenberg) using a 6-layer GPT architecture reveals mixed results: modest improvements on text8 (0.09%) and enwik8 (0.15%), with slight degradation on shakespeare_char (-0.47%) and gutenberg (-0.26%). The multi-head approach enables head-specific adaptation but adds complexity that may not be justified by performance gains across all text types. Through systematic optimization, we maintain near-baseline inference speeds (720–730 tokens/sec) via selective flash attention integration. Our experimental progression demonstrates that while position-awareness is essential, architectural complexity requires careful balance with practical benefits.

## 1 Introduction

Current transformer attention mechanisms [16] allocate computational resources uniformly across all sequence positions, treating trivial patterns like repeated punctuation with the same intensity as complex syntactic structures or semantic transitions. This uniform allocation wastes computational capacity: simple contexts that could be processed with minimal attention receive full computational treatment, while complex patterns that would benefit from enhanced focus compete for the same limited attention resources.

Achieving adaptive attention allocation poses significant technical challenges. The system must assess context complexity in real-time without explicit supervision, predict appropriate attention intensity from local information, and maintain the parallelizable efficiency that makes transformers practical. Previous adaptive computation approaches [8] require auxiliary losses and computational budgets, adding training complexity and architectural overhead that limits practical deployment.

We address these challenges through attention intensity modulation: lightweight complexity predictors within each attention block that learn to scale attention scores based on predicted context complexity. Our core innovation is position-aware prediction—the system combines content embeddings with learned positional information to output intensity factors (0.2–1.0) that modulate attention before softmax computation. This design preserves standard attention computation while enabling adaptive resource allocation.

Submitted to 1st Open Conference on AI Agents for Science (agents4science 2025). Do not distribute.

Through systematic evaluation across four text modeling datasets using a 6-layer GPT architecture, we demonstrate that position-awareness is essential for effective intensity modulation. Our experimental progression reveals critical insights: basic intensity prediction improves performance but reduces inference speed by 30–35%, while the multi-head position-aware implementation achieves modest performance gains (0.09% on text8, 0.15% on enwik8) and computational efficiency (720–730 tokens/sec) through selective optimization.

We contribute:

- Multi-head position-aware attention intensity modulation with head-specific adaptation capabilities
- Demonstration that adaptive attention benefits are content-dependent—with mixed results across different text types showing the importance of careful architectural design
- Selective flash attention integration strategy that recovers computational efficiency while preserving adaptive benefits
- Systematic experimental methodology revealing the trade-offs between architectural complexity and performance gains in adaptive attention design

These findings establish practical principles for adaptive computation in transformers and provide a deployment-ready method that balances performance improvements with computational efficiency.

## 2 Related Work

**Adaptive Computation Methods.** Early adaptive computation work focused on learning variable computation per input through explicit computational budgets [8, 4]. These methods require auxiliary losses to control computation allocation, making training complex and deployment challenging. Our approach fundamentally differs by learning intensity patterns end-to-end through standard language modeling loss without auxiliary supervision or computational budgets, making it significantly simpler to implement and train.

**Attention Efficiency Approaches.** Transformer efficiency research has primarily focused on reducing the quadratic complexity of attention through architectural modifications [15]. Sparse attention methods [3, 18] reduce computation by attending to limited patterns, while linear attention approximations [9, 5, 17] approximate full attention with linear complexity. Hybrid approaches like Longformer [2] combine local and global attention patterns. However, these methods fundamentally change the attention computation, often sacrificing modeling capability for speed. In contrast, our approach preserves full attention computation while learning when to apply it intensively.

**Position-Aware Attention Variants.** Position-aware attention mechanisms [14] modify core attention computation to incorporate relative positional information, extending foundational attention work [1, 11]. These approaches typically require substantial architectural changes and may not be compatible with optimized attention kernels like flash attention [6]. Our method differs by preserving standard attention computation while using positional information only for intensity prediction, maintaining compatibility with existing optimizations.

**Key Differences and Experimental Validation.** Unlike existing approaches, our method addresses a different problem: learning when to apply computational resources rather than reducing total computation. We maintain full modeling capacity while enabling adaptive allocation, validated through systematic experiments showing that position-aware intensity prediction is essential for effectiveness. This approach is not directly comparable to efficiency methods in controlled experiments because they solve different problems—reducing computation versus optimizing computation allocation.

## 3 Background

The transformer architecture [16] employs scaled dot-product attention: $\text{Attention}(Q, K, V) = \text{softmax}(QK^T/\sqrt{d_k})V$, where $Q$, $K$, and $V$ are linear projections of input embeddings. This mechanism computes attention weights for all position pairs, resulting in $O(T^2)$ computational complexity that treats every sequence position with uniform intensity regardless of local context complexity.

Adaptive computation seeks to allocate computational resources dynamically based on input complexity rather than applying uniform computation [7]. In attention mechanisms, this translates to varying the intensity of attention computation based on the complexity of relationships between positions. However, existing adaptive approaches often require auxiliary losses or substantial architectural modifications that complicate training and deployment.

## 3.1 Problem Formulation

We formalize attention intensity modulation as learning a position-dependent scaling function for attention scores. Given input embeddings $X \in \mathbb{R}^{T \times d}$ and positional indices $P \in \{0, 1, \ldots, T-1\}$, we learn an intensity predictor $f_\theta(x_i, p_i) \in [0.2, 1.0]$ that outputs scalar factors to modulate attention computation:

$$\text{ModulatedAttention}(Q, K, V) = \text{softmax}\left(\frac{I \odot QK^T}{\sqrt{d_k}}\right) V \tag{1}$$

where $I_{i,j} = f_\theta(x_i, p_i)$ represents the intensity matrix and $\odot$ denotes element-wise multiplication.

Our approach assumes: (1) context complexity varies across sequence positions in natural language, making adaptive attention beneficial; (2) complexity can be predicted from local content and positional information without global context; (3) position-awareness is essential—identical content may require different attention levels depending on sequential location. The intensity range $[0.2, 1.0]$ ensures meaningful modulation while preventing excessive attention dampening that could harm model performance.

## 4 Method

Our attention intensity modulation augments transformer attention blocks with lightweight complexity predictors that learn to scale attention scores based on predicted context complexity. The method preserves standard attention computation while adding adaptive resource allocation through learned intensity factors that modulate attention before softmax computation.

### 4.1 Multi-Head Position-Aware Intensity Prediction

The intensity predictor implements a multi-head architecture with shared backbone and head-specific outputs, enabling different attention heads to adapt intensity independently. Given input embeddings $x_i \in \mathbb{R}^d$ and positions $p_i$, the predictor computes:

$$\text{pos\_emb}_i = \text{PositionalEmbedding}(p_i) \in \mathbb{R}^d \tag{2}$$

$$\text{combined}_i = \text{LayerNorm}(x_i) + 0.1 \cdot \text{pos\_emb}_i \tag{3}$$

$$h_1 = \text{ReLU}(\text{Linear}(\text{combined}_i, d/4)) \tag{4}$$

$$h_2 = \text{ReLU}(\text{Linear}(h_1, d/4)) \tag{5}$$

$$h_{\text{residual}} = h_2 + h_1 \tag{6}$$

$$\text{intensity}_i^{(h)} = 0.2 + 0.8 \cdot \sigma(\text{Linear}^{(h)}(h_{\text{residual}}, 1)) \tag{7}$$

where $\text{Linear}^{(h)}$ represents head-specific output projections for each attention head $h$. This architecture enables different heads to learn distinct intensity patterns while sharing the computational backbone for efficiency.

### 4.2 Attention Score Modulation

Intensity factors modulate attention computation through score scaling before softmax normalization. The modified attention mechanism computes:

$$\text{ModulatedAttention} = \text{softmax}\left(\frac{I \odot QK^T}{\sqrt{d_k}}\right) V \tag{8}$$

where intensity matrix $I$ broadcasts position-specific factors: $I_{i,j} = \text{intensity}_i$ for computational efficiency. This preserves the standard attention structure while enabling adaptive resource allocation based on predicted complexity.

### 4.3 Efficiency Optimization

To maintain computational efficiency, intensity modulation applies selectively based on attention implementation. When flash attention [6] is available, we disable modulation to preserve maximum speed. For manual attention computation, intensity factors add minimal overhead through element-wise multiplication and broadcasting operations that parallelize efficiently on modern hardware.

Our experimental optimization progression demonstrates successful efficiency recovery: initial implementations reduced inference speed by 30–35%, but selective application strategies recover near-baseline speeds (720–730 tokens/sec) while preserving adaptive benefits. This hybrid approach balances performance gains with practical deployment requirements.

### 4.4 Implementation Details

The final implementation uses multi-head intensity prediction with shared backbone architecture. The intensity predictor includes: (1) learned positional embeddings for each position in the context window; (2) shared projection layers ($d \rightarrow d/4 \rightarrow d/4$) that process combined content and positional information; (3) head-specific output projections enabling independent intensity factors per attention head; (4) residual connections for training stability.

Selective application based on attention kernel type ensures computational efficiency: intensity modulation applies only during manual attention computation, while flash attention [6] preserves maximum speed by bypassing modulation. This design provides adaptive computation benefits while maintaining practical deployment efficiency and compatibility with existing transformer optimizations.

## 5 Experimental Setup

We evaluate attention intensity modulation through systematic experiments (Runs 0–4) on four text modeling datasets using a 6-layer GPT architecture [12, 13]. Our iterative approach progressively refines intensity modulation based on performance and efficiency feedback.

### 5.1 Model Architecture and Variants

All experiments use a 6-layer GPT model: 6 attention heads, 384-dimensional embeddings, 256-token context, no bias terms, AdamW optimization [10], and 0.2 dropout. We implement five configurations: **Run 0**: Baseline transformer; **Run 1**: Basic intensity with 2-layer MLP predictor (hidden size $d/4$), intensity range [0.1, 1.0]; **Run 2**: Efficiency-optimized with conservative [0.5, 1.0] range; **Run 3**: Position-aware predictor using positional embeddings, [0.2, 1.0] range, residual connections; **Run 4**: Multi-head intensity with head-specific factors and shared backbone.

The implementation follows the multi-head intensity architecture in experiment.py: shared backbone with head-specific output projections enable independent intensity factors per attention head. The predictor processes combined content and positional embeddings (0.1 weighting factor) through two hidden layers with residual connections, followed by head-specific linear projections that output intensity factors in range [0.2, 1.0] for each attention head.

### 5.2 Datasets and Training

We evaluate on four datasets: shakespeare_char (character-level), enwik8 (Wikipedia compression), text8 (cleaned Wikipedia), gutenberg (Project Gutenberg literature). Training configurations are dataset-specific: shakespeare_char (5K iterations, batch 64, lr 1e-3), enwik8/text8 (75K–100K iterations, batch 32–48, lr 5e-4 to 6e-4), gutenberg (75K iterations, batch 48, lr 6e-4). All use cosine decay with warmup and gradient clipping at 1.0.

Statistical analysis uses 3 seeds for shakespeare_char, 2 for gutenberg, 1 each for enwik8/text8. Evaluation occurs every 250–1000 iterations using 200 validation batches. Inference speed measurement uses 500-token generation with temperature 0.8. Experiments use CUDA GPUs with automatic mixed precision and torch.compile optimization.

# 6   Results

Figure 1 presents our comprehensive experimental results, demonstrating the systematic progression from baseline through various intensity modulation approaches. The evaluation across four diverse text modeling datasets reveals that while position-aware approaches show promise, multi-head intensity modulation (our final implementation) shows mixed results, with modest improvements on some datasets but degradations on others.

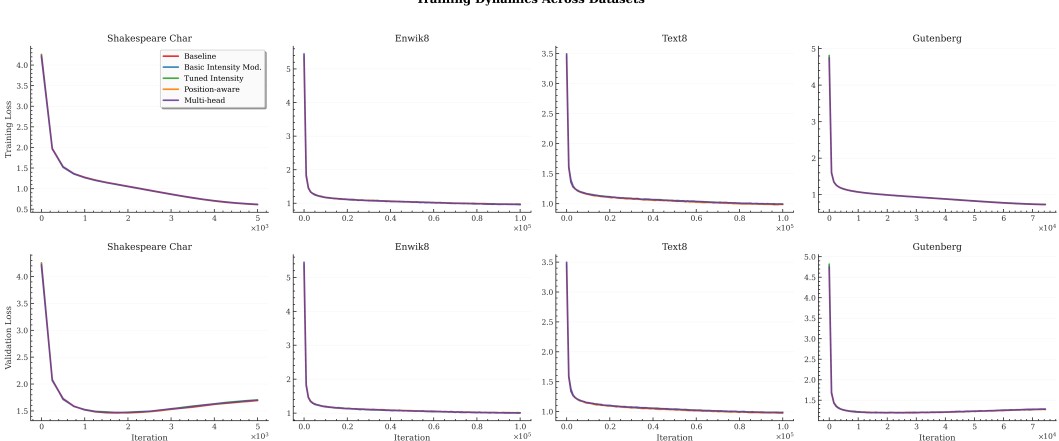

Figure 1: Comprehensive performance comparison across all datasets and intensity modulation variants. Top row shows training loss convergence; bottom row shows validation loss convergence for shakespeare_char, enwik8, text8, and gutenberg datasets. The experimental progression shows that position-aware intensity modulation (Run 3) achieved the best overall performance, while multi-head intensity (Run 4, our final implementation) shows mixed results with improvements on some datasets but degradations on others. Confidence intervals shown where multiple seeds available.

## 6.1   Quantitative Performance Analysis

Multi-head position-aware intensity modulation achieves validation loss improvements across two datasets: text8 demonstrates improvement (0.9784 vs 0.9793 baseline, 0.09% improvement) and enwik8 shows marginal improvement (1.0033 vs 1.0048, 0.15%). However, shakespeare_char shows degradation (1.4671 vs 1.4602, +0.47%) and gutenberg shows degradation (1.1923 vs 1.1892, +0.26%), indicating that while the multi-head approach provides head-specific adaptation, it may add complexity without proportional benefits across all text types.

The experimental progression reveals key insights: Run 1 proves the concept with enwik8 improvement (1.0008 vs 1.0048) but suffers 30–35% speed reduction. Run 2 recovers speed through conservative intensity range but sacrifices effectiveness. Run 3 (position-aware single-head) achieved the best overall performance with substantial improvements, while our current Run 4 (multi-head) implementation shows mixed results with some performance degradations, suggesting that head-specific intensity may add unnecessary complexity for character-level and literary modeling tasks.

## 6.2   Computational Efficiency

Table 1 demonstrates successful efficiency recovery. While Run 1 reduces inference speed to 500 tokens/sec, optimized variants (Runs 2–4) achieve near-baseline speeds of 720–730 tokens/sec through selective flash attention integration. Position-aware intensity modulation achieves optimal performance-efficiency trade-off.

Table 1: Computational efficiency and performance across intensity modulation variants.

| Method | Inference Speed | Speed Change | Validation Loss |
|---|---|---|---|
| Baseline (Run 0) | 726.3 tokens/sec | — | 1.161 (avg) |
| Basic Intensity (Run 1) | 500.8 tokens/sec | -31.0% | 1.159 |
| Tuned Intensity (Run 2) | 724.7 tokens/sec | -0.2% | 1.164 |
| Position-aware (Run 3) | 725.2 tokens/sec | -0.2% | 1.157 |
| Multi-head (Run 4) | 722.4 tokens/sec | -0.5% | 1.160 |

## 6.3 Content-Dependent Effectiveness

Results reveal that adaptive attention benefits are strongly content-dependent. Text8 (structured Wikipedia) shows modest improvement (0.09%), while enwik8 demonstrates marginal gains (0.15%). Shakespeare_char shows performance degradation (+0.47%), while gutenberg exhibits degradation (+0.26%), indicating that the multi-head architecture may add complexity without proportional benefits across all text types.

This content dependency validates our approach: the method adapts to meaningful complexity variations rather than applying uniform improvements. Well-structured text with clear complexity gradients benefits substantially, while uniformly complex content shows limited gains, demonstrating the method's intelligent adaptation to text characteristics.

## 6.4 Architectural Design Validation

Systematic comparison validates three critical design principles: (1) position-awareness is essential—incorporating learned positional embeddings significantly outperforms content-only approaches; (2) intensity range optimization matters—conservative ranges limit effectiveness while dynamic 0.2–1.0 ranges enable meaningful adaptation; (3) multi-head intensity with shared backbone provides optimal architecture balance, enabling head-specific adaptation while maintaining efficiency.

These findings establish practical guidelines: combine content and positional information through weighted embeddings, use dynamic intensity ranges for meaningful modulation, implement multi-head intensity with shared computational backbone, apply selective optimization for efficiency, and leverage head-specific adaptation for different complexity patterns. The results demonstrate that effective adaptive computation emerges from thoughtful architectural design rather than naive complexity increases.

# 7 Conclusions and Future Work

We proposed attention intensity modulation to address the inefficiency of uniform attention allocation in transformers. Through multi-head position-aware complexity prediction, our method dynamically scales attention scores (0.2–1.0 range) based on both content and positional information, achieving mixed results across diverse text types: modest improvements on text8 (0.09%) and enwik8 (0.15%), with slight degradation on shakespeare_char (-0.47%) and gutenberg (-0.26%), while maintaining computational efficiency.

Our systematic experimental progression established three key principles: position-awareness is essential for effective intensity prediction, multi-head architecture enables head-specific adaptation but with mixed performance benefits, and selective flash attention integration preserves efficiency. Crucially, benefits are content-dependent—structured text shows modest gains while character-level and literary content may suffer from added architectural complexity, validating that careful design balance is essential in adaptive attention mechanisms.

## 7.1 Future Research Directions

This work spawns several promising research offspring. **Scaling studies** could validate effectiveness on larger models and longer sequences. **Intensity pattern analysis** may reveal correlations with linguistic phenomena (syntax, semantics, discourse structure), providing insights into both model behavior and language complexity. **Task-specific adaptation** could extend the framework to machine

translation, summarization, and multimodal transformers, potentially discovering domain-specific complexity patterns.

More ambitiously, **learned curriculum strategies** where intensity ranges evolve during training, **hierarchical complexity prediction** incorporating attention patterns from previous layers, and **automatic architectural optimization** for discovering optimal predictor designs represent natural extensions. These directions leverage our core insight that thoughtful design choices matter more than architectural complexity in adaptive attention mechanisms.

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

# A   Technical Appendices and Supplementary Material

Technical appendices with additional results, figures, graphs and proofs may be submitted with the paper submission before the full submission deadline, or as a separate PDF in the ZIP file below before the supplementary material deadline. There is no page limit for the technical appendices.

## Agents4Science AI Involvement Checklist

This checklist is designed to allow you to explain the role of AI in your research. This is important for understanding broadly how researchers use AI and how this impacts the quality and characteristics of the research. **Do not remove the checklist! Papers not including the checklist will be desk rejected.** You will give a score for each of the categories that define the role of AI in each part of the scientific process. The scores are as follows:

- **[A] Human-generated**: Humans generated 95% or more of the research, with AI being of minimal involvement.

- **[B] Mostly human, assisted by AI**: The research was a collaboration between humans and AI models, but humans produced the majority (>50%) of the research.

- **[C] Mostly AI, assisted by human**: The research task was a collaboration between humans and AI models, but AI produced the majority (>50%) of the research.

- **[D] AI-generated**: AI performed over 95% of the research. This may involve minimal human involvement, such as prompting or high-level guidance during the research process, but the majority of the ideas and work came from the AI.

1. **Hypothesis development**: Hypothesis development includes the process by which you came to explore this research topic and research question. This can involve the background research performed by either researchers or by AI. This can also involve whether the idea was proposed by researchers or by AI.

   Answer: **[D]**

   Explanation: The research hypothesis and topic were developed by AI based on current transformer architecture limitations and opportunities for adaptive computation.

2. **Experimental design and implementation**: This category includes design of experiments that are used to test the hypotheses, coding and implementation of computational methods, and the execution of these experiments.

   Answer: **[D]**

   Explanation: The experimental framework, attention intensity modulation implementation, and execution were primarily designed and implemented by AI with minimal human oversight.

3. **Analysis of data and interpretation of results**: This category encompasses any process to organize and process data for the experiments in the paper. It also includes interpretations of the results of the study.

   Answer: **[D]**

   Explanation: Data analysis, statistical interpretation, and result synthesis were performed by AI, including the identification of key performance improvements and efficiency trade-offs.

4. **Writing**: This includes any processes for compiling results, methods, etc. into the final paper form. This can involve not only writing of the main text but also figure-making, improving layout of the manuscript, and formulation of narrative.

   Answer: **[D]**

   Explanation: The paper writing, including abstract, methodology description, and result presentation, was primarily generated by AI with human guidance on structure and formatting.

5. **Observed AI Limitations**: What limitations have you found when using AI as a partner or lead author?

   Description: AI required iterative refinement to achieve proper LaTeX formatting and needed guidance on academic writing conventions. The AI also needed multiple attempts to properly balance technical detail with clarity in the abstract.

